## RESEARCH ARTICLE

# A conserved *C. elegans* zinc finger homeodomain protein, ZFH-2, is continuously required for the structural integrity and function of the alimentary tract and gonad

Antoine Sussfeld*, Berta Vidal*, Surojit Sural*, Daniel M. Merritt, G. Robert Aguilar, Yasmin H. Ramadan and Oliver Hobert‡

## ABSTRACT

An unusually large transcription factor arose at the base of bilaterian evolution through domain shuffling that recombined many copies of two distinct DNA-binding domains, C2H2-type zinc fingers and homeodomains. The function of this deeply conserved type of protein remains poorly characterized. We describe here the complete and complex expression pattern of its sole *Caenorhabditis elegans* representative, ZFH-2, throughout development and adulthood. We show that animals lacking this protein display defects in proper alimentary tract formation and starve to death in the first larval stage with an apparent inability to ingest food. Conditional removal of ZFH-2 at post-developmental stages reveals a continuous function of this protein in enabling food ingestion and demonstrates additional essential functions for the formation of other, post-embryonically generated tubular structures. Even though ZFH-2 is broadly expressed throughout the nervous system, we detected no obvious defects in neuronal development or function in *zfh-2* null mutants. Genome-engineered alleles indicate that, although a large part of the protein is dispensable, at least a subset of the homeodomains are critical determinants for the essential functions of this protein.

KEY WORDS: *C. elegans*, Homeobox gene, Development

## INTRODUCTION

The existence of homeodomain transcription factors dates back to unicellular eukaryotes (Larroux et al., 2008; Sebe-Pedros et al., 2011). With the advent of animal multicellularity, homeodomain-encoding homeobox genes multiplied and some also acquired additional domains, such as the LIM domain or CUT domain, through the process of domain shuffling, a major driver of evolutionary novelty (Chothia et al., 2003). Another domain shuffling event occurred later, at the base of bilaterian evolution, through the recombination of multiple C2H2 zinc-finger domains with one or multiple homeodomains. These shuffling events resulted in two types of zinc finger homeodomain proteins that have remained evolutionary stable throughout all bilaterian phyla but are absent in more basal metazoans, such as Cnidaria or sponges (Brauchle et al., 2018; Burglin and Affolter, 2016). One type is characterized by a single, centrally located homeodomain, surrounded by multiple (but always fewer than ten) C2H2 zinc fingers (Fig. 1A). Members of this type of zinc finger homeodomain protein are called ZEB1 and ZEB2 (also known as ZFHX1a and ZFHX1b) in vertebrates, Zfh1 in *Drosophila* and ZAG-1 in *C. elegans* (Fig. 1A) (Burglin and Affolter, 2016; Ruvkun and Hobert, 1998; Vandewalle et al., 2009). Several reports have characterized the function of the ZEB family across animal phylogeny using null mutant approaches in mice, flies and worms, identifying, among other functions, crucial roles in mesoderm and nervous system development as well as during tumorigenesis (reviewed by Vandewalle et al., 2009).

The second, independent C2H2 zinc finger and homeodomain domain shuffling and fusion event that occurred at the base of bilaterian evolution created an even larger type of zinc finger homeodomain protein (Fig. 1B) (Burglin and Affolter, 2016). Across all bilaterian phyla, these very large proteins are composed of more than 1000 amino acids and contain three to four homeodomains, interspersed in a characteristic pattern with up to more than 20 C2H2 zinc fingers (Fig. 1B). Members of this type of protein include three vertebrate homologs (ZFHX2, ZFHX3 and ZFHX4), a single echinoderm, mollusk and arthropod homolog, as well as a single *C. elegans* homolog, called ZFH-2 (Fig. 1B).

Even though not comprehensively investigated yet, there are several hints pointing towards the importance of ZFHX2, ZFHX3 and ZFHX4 in various contexts. Mouse ZFHX2 mutants are viable but display several behavioral deficits (Komine et al., 2012), while mouse ZFHX3 (also known as ATBF1) is essential for viability (Sun et al., 2012) and has been implicated in tumor formation (Sun et al., 2005) as well as regulation of circadian rhythms (Parsons et al., 2015). ZFHX4 mutant mice die perinatally due to respiratory problems (Zhang et al., 2021). ZFHX2, ZFHX3 and ZFHX4 variants have also been associated with distinct human neurological disorders (Habib et al., 2018; Perez Baca et al., 2024, 2025), but the cellular focus of action of these genes in the brain, as well as their mechanism of action, is poorly understood. Moreover, owing to the size of the respective genetic loci, no complete locus deletions of these genes have been generated, leaving the null phenotype of these genes unclear. Through the use of P-element insertion and RNAi-mediated knockdown, the function of the single *Drosophila* homolog of ZFHX2/3/4, called Zfh2, has been described in very specific developmental contexts within the nervous system and during wing and leg patterning (Gabilondo et al., 2011; Guarner et al., 2014; Guntur et al., 2023; Rojas Villa et al., 2019; Whitworth and Russell, 2003). Again, it remains unresolved whether these alleles are complete null alleles.

Department of Biological Sciences, Columbia University, Howard Hughes Medical Institute, New York, NY 10027, USA.
*These authors contributed equally to this work

‡Author for correspondence (or38@columbia.edu)

S.S., 0000-0002-0422-9799; D.M.M., 0000-0001-6273-6644; G.R.A., 0000-0001-6926-0319; O.H., 0000-0002-7634-2854

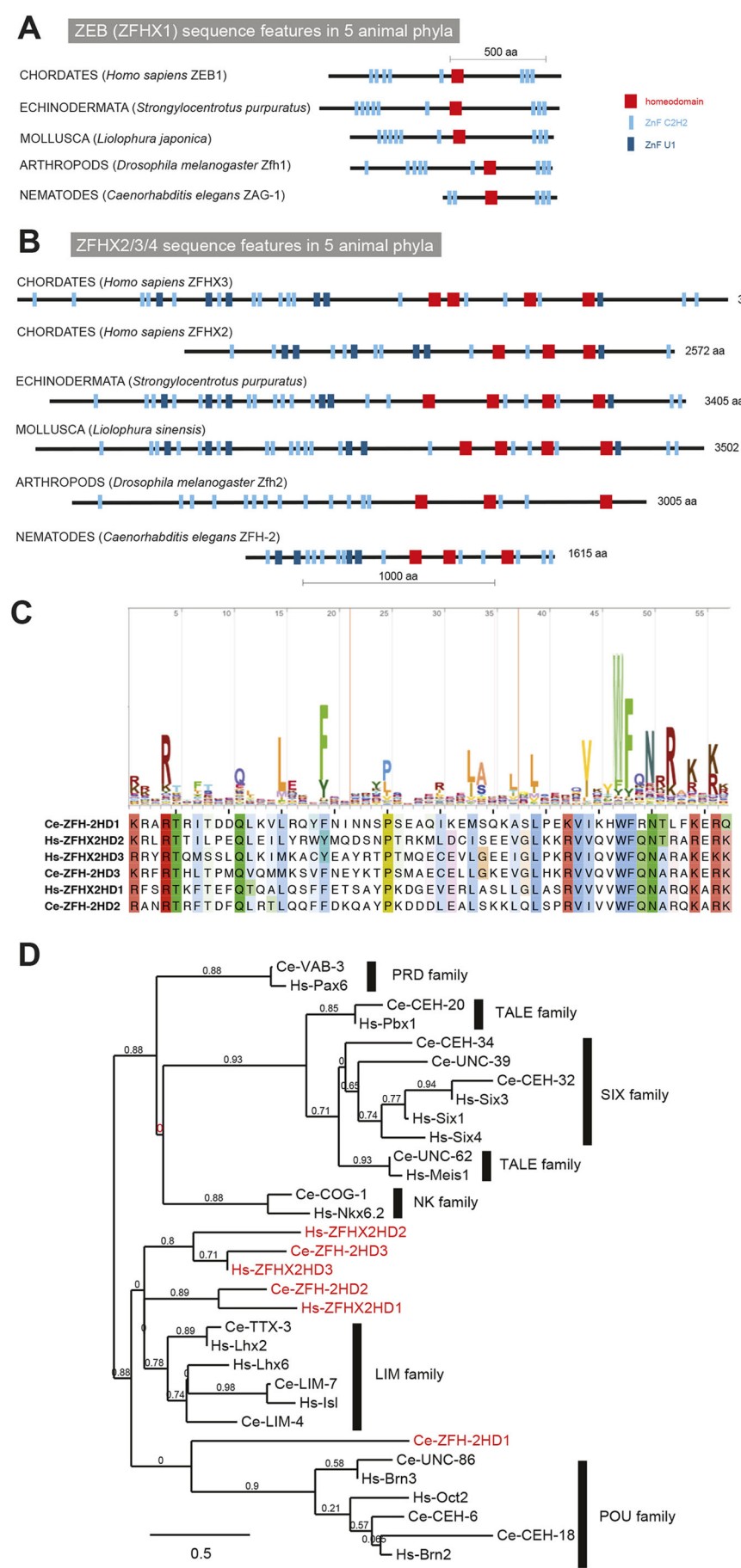

**Fig. 1. Overview of zinc finger homeodomain proteins.** (A,B) Domain structure of ZEB/ZFHX1 subtype (A) and ZFHX2/3/4 subtype (B) of zinc finger homeodomain proteins from several distinct animal phyla. Only two representative chordate proteins are shown. (C) Alignments of homeodomains in Ce ZFH-2 and Hs ZFHX2 proteins with the consensus sequence for homeodomains (PFAM Logo: PF00046). Alignment was generated using Clustal Omega: https://www.ebi.ac.uk/jdispatcher/msa/clustalo. Colors in the alignment represent amino acid similarity between sequences in Clustal X scheme. Ce, *C. elegans*; Hs, *Homo sapiens*. (D) Relationship of homeodomains in ZFHX2/3/4 proteins (red) with that of other homeodomains. This tree was built at www.phylogeny.fr (Dereeper et al., 2008) with default parameters.

The single *C. elegans* homolog of ZFHX2/3/4 has not been studied in any detail before. The cataloguing of many essential genes on chromosome I has isolated, among other loci, a premature stop codon in *zfh-2* (Chu et al., 2014), but the timing or cause of lethality has not been reported. An RNAi-based screen for fat storage mutants has revealed several genes, among them *zfh-2*, for which depletion results in increased intestinal fat staining (Ke et al., 2021), but this RNAi phenotype has not been independently validated. Like its fly and vertebrate homologs, *C. elegans* ZFH-2 is expressed in many neurons of the mature nervous system, as revealed in a survey of homeodomain protein expression patterns (Reilly et al., 2020), but ZFH-2 protein expression outside the nervous system, or the function within the nervous system, has not previously been examined. Since many homeodomain proteins have been found to play prominent roles in nervous system development (Hobert, 2021) and since this specific zinc finger homeobox gene family specifically arose at the base of bilaterian evolution, our expectation going into studying *zfh-2* function in *C. elegans* has been to uncover fundamental functions of this gene in nervous system development.

Here, we undertake a comprehensive expression pattern as well as mutant analysis of the ZFH-2 protein. Using a CRISPR/Cas9-engineered reporter allele, we define broad, but nevertheless highly cell type-specific expression in multiple distinct tissue types. We characterize the organismal effect of complete removal of the *zfh-2* locus, as well as the conditional, post-developmental removal of ZFH-2 protein and we undertake a structure/function analysis of the gene locus to probe the function of individual isoforms and domains of this large protein. Our results reveal essential roles of this gene in several distinct organs, but, surprisingly, no apparent function in nervous system development.

## RESULTS
### ZFH-2 is the sole *C. elegans* representative of the ZFHX2/3/4 subtype of zinc finger homeodomain proteins
Based on both reciprocal BLAST searches, DIOPT scores and overall domain organization, the *C. elegans* ZFH-2 protein is the sole ortholog of the ZFHX2/3/4 family of zinc finger homeodomain proteins (Fig. 1B) (Ruvkun and Hobert, 1998). Missing several C2H2 zinc fingers, the 1615 amino acid *C. elegans* ZFH-2 protein is shorter than orthologs from other phyla (e.g. human ZFHX2 is 2572 amino acids long), but the overall arrangement of homeodomains and zinc fingers is similar to that of orthologs in other species (Fig. 1B), corroborating the origin from a common ancestor.

Like its vertebrate homologs, the C2H2 zinc fingers of ZFH-2 come in two flavors, conventional C2H2 zinc fingers usually associated with DNA binding (SMART domain SM000355), but also a few U1-type C2H2 zinc fingers, usually associated with RNA binding (SMART domain SM00451). While RNA binding has not yet been reported for zinc finger homeodomain proteins, a diversity of distinct binding partners (i.e. DNA and RNA) is an attractive possibility for a protein with so many nucleic acid-binding domains.

An alignment with the consensus sequence for homeodomains (PFAM Logo: PF00046) shows that all ZFH-2 homeodomains match the DNA-binding consensus sequence (Fig. 1C). However, some of the homeodomains of *C. elegans* ZFH-2 are more similar to vertebrate ZFHX orthologs than others (Fig. 1D).

### Expression pattern of a GFP-tagged ZFH-2 protein
Using a CRISPR/Cas9-genome engineered reporter allele, we had previously described the expression of ZFH-2::GFP in the mature nervous system of the worm, revealing broad expression in 54 of all

118 neuron classes, spanning all major ganglia (Reilly et al., 2020) (Table S1). Within specific regions of the nervous system, ZFH-2 is restricted. For example, it is expressed in all cholinergic, but no GABAergic, motor neurons in the ventral nerve cord and only in a single class of the 14 pharyngeal enteric neurons.

We re-examined this reporter allele, but this time removed a floxed *unc-119* selectable marker cassette introduced downstream of the locus (see Materials and Methods). This removal had no obvious impact on expression in the nervous system. Neuronal expression after hatching and during larval stages appears to match expression that we previously described in the L4/adult nervous system, with the exception of the CAN neuron in which we now detect ZFH-2::GFP expression (Fig. 2B).

By assessing overlap of expression with the pan-glia marker *mir-228* (Pierce et al., 2008), we found that ZFH-2::GFP is expressed in all ectodermal glial cells (sheath and socket glia) in the hermaphrodite (Fig. S1A). Outside the nervous system, ZFH-2::GFP protein is expressed in multiple cell types, yet again in a selective manner, as summarized in Table S1. In the pharynx, expression was observed in a subset of muscle and epithelial cells. Expression was also observed in pharyngeal-intestinal valve (vpi) cells, a group of six equivalent interlocking cells that link the posterior bulb of the pharynx to the anterior four cells of the intestine. These six cells comprise a small epithelial channel linking the lumen of the pharynx to the large lumen of the anterior intestine (Hall and Altun, 2007). Within the midgut, ZFH-2 expression was observed only in the posterior-most intestinal cells (Fig. 2B). ZFH-2 was prominently expressed throughout the hindgut, including the rectal valve and gland cells and all rectal epithelial cells (Fig. 2B). Another tubular set of structures that express ZFH-2 is the excretory system, as well as gonadal structures. In the excretory system, ZFH-2 was expressed throughout all its constituent cell types (canal, pore, duct, gland and canal-associated CAN neuron). In the gonad, ZFH-2 was expressed in gonadal sheath cells, spermatheca and uterine cells, as well as in what appear to be the first two circular rows of vulval cells (VulE and VulF) (Fig. 2B). No expression was observed in body wall muscle or hypodermal cells along the length of the animal.

In the embryo, we observed ZFH-2::GFP expression to commence at around the bean stage (Fig. 2B), at about the time many cells exit the cell cycle to terminally differentiate. This observation is consistent with a previous 4D lineage expression analysis of many reporter-tagged transcription factors, including ZFH-2 (Ma et al., 2021). The pattern of ZFH-2::GFP expression right after hatching appears to be similar to that of an adult animal, except for the cell and tissue types that are formed only later in larval development (e.g. gonad and vulva). Taken together, our analysis of the endogenously tagged ZFH-2 protein refines, validates and extends previously reported expression data, including single-cell RNA-sequencing studies (Ghaddar et al., 2023; Packer et al., 2019; Taylor et al., 2021).

### A null allele of the *zfh-2* locus causes larval arrest
A previous screen for essential *C. elegans* genes in a short interval on chromosome I, covered by a free duplication, identified a nonsense allele in *zfh-2*, *h379* (Chu et al., 2014). This allele introduces a premature stop codon in exon 12, between zinc fingers 10 and 11, before the first homeodomain (Fig. 2A), but these animals have not been examined for timing or cause of death, and they are no longer available. The *C. elegans* knockout group at Tokyo Women's Medical University Hospital has isolated a 1.6 kb deletion allele, *tm310*, that starts after zinc finger 10 and terminates in an intron located within the first homeobox (Fig. 2A), possibly

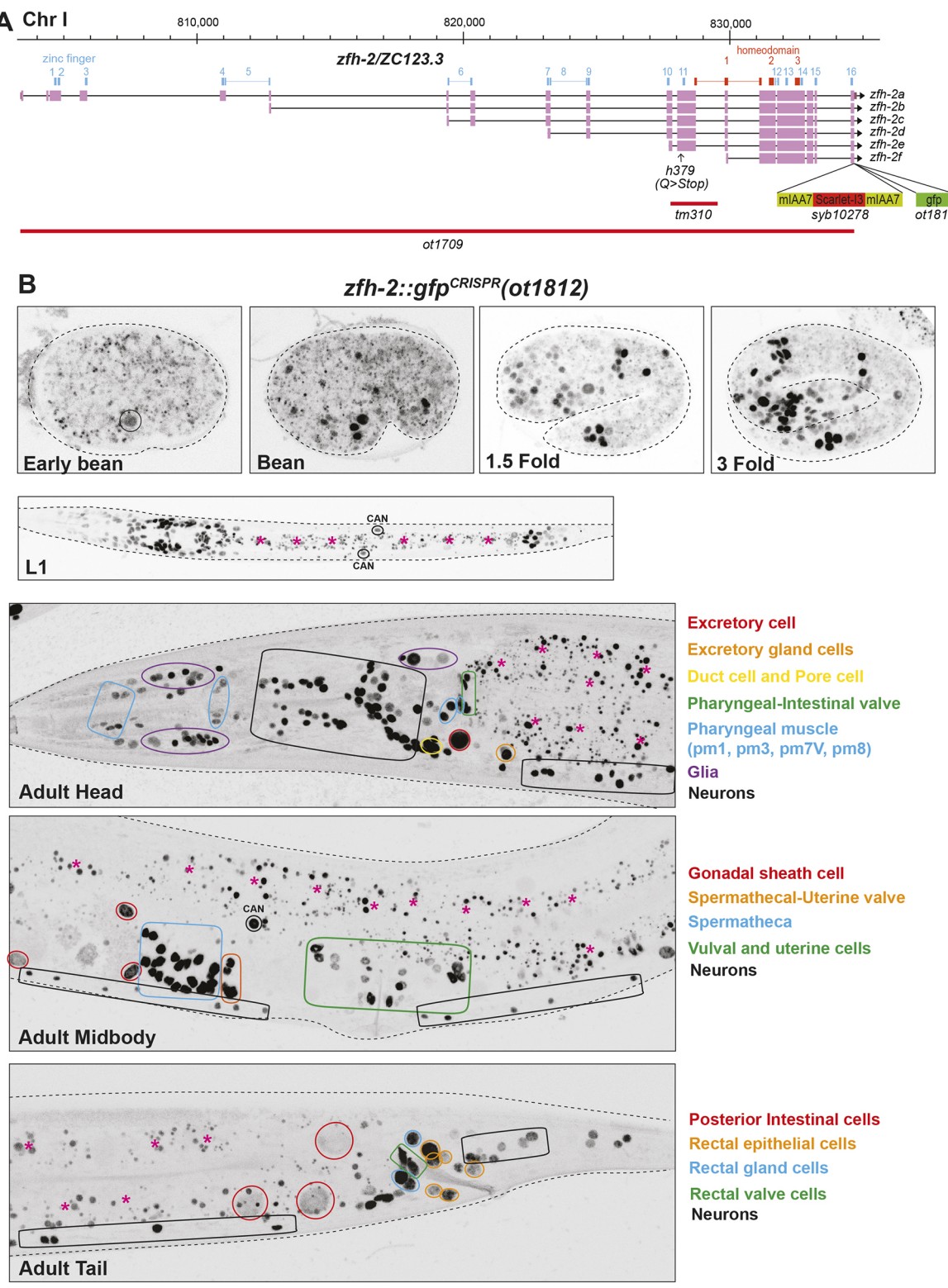

**Fig. 2. _zfh-2_ locus, alleles and expression patterns.** (A) _zfh-2_ locus showing a null allele (_ot1709_) generated via the CRISPR/Cas9 system and two reporter alleles, one containing the mIAA7 degrons and an _mScarlet-I3_ reporter (_syb10278_) and the other containing a _gfp_ reporter (_ot1812_). The latter (_ot1812_) was generated using the modEncode-generated _zfh-2(st12167[zfh-2::GFP+loxP+unc-119(+)+loxP])I_ allele and excising the _unc-119(+)_ cassette with germline Cre recombinase. (B) Expression of _zfh-2(ot1812)_ reporter allele over the course of development. Expression of ZFH-2 in non-neuronal tissues is labeled in different colors, as indicated. Gut autofluorescence is indicated with pink asterisks. Dashed lines delineate outline of worms. Neuronal expression is listed in Table S1.

resulting in a premature termination of protein production. Animals carrying this allele were catalogued as being either lethal or sterile (https://shigen.nig.ac.jp/c.elegans/DetailsSearch?allele=tm310).

We set out to generate an unambiguous molecular null allele of _zfh-2_. We used the CRISPR/Cas9 system to engineer a >30 kb deletion of the entire _zfh-2_ locus (Fig. 2A). We balanced this

deletion, *ot1709*, over the aneuploidy-free balancer *tmC20* (Dejima et al., 2018).

Homozygous *zfh-2(ot1709)* null mutant animals showed a completely penetrant first larval stage arrest phenotype (Fig. 3A). Directly after hatching, these animals appeared morphologically wild type and were able to move around, but soon acquired a scrawny morphology, with a wrinkled, deflated appearance of the intestinal lumen (Fig. 3B). The size, scrawny morphology and arrest phenotype is similar to that of wild-type animals hatching in the absence of food, indicating possible feeding defects of *zfh-2* mutant animals. We assessed the ability of animals to take in fluorescently labeled beads and found that, in contrast to wild-type animals, *zfh-2* mutants cannot ingest such particles (Fig. 3C). These phenotypes are the apparent result of an inability of the animals to engage in pharyngeal (foregut) pumping (Fig. 3D). The NSM neurons, which are the only neurons in the pharyngeal circuit that express *zfh-2* (Table S1) (Reilly et al., 2020), detect food signals in the pharyngeal lumen and increase the rate of pumping via releasing serotonin (Estrem et al., 2025 preprint). We found that the strong pharyngeal pumping defects in *zfh-2* mutant animals persisted even after the addition of exogenous serotonin (Fig. S2), a key inducer of pharyngeal pumping that acts directly on the pacemaker MC neurons to promote muscle contractions (Song and Avery, 2012). This indicates a function of *zfh-2* that is independent of the serotonergic NSM neurons.

The expression of *zfh-2* in pharyngeal muscle prompted us to examine pharyngeal muscle differentiation. Using a reporter transgene for the myosin-encoding *myo-2* gene, we indeed find partially penetrant defects in pharyngeal muscle differentiation in *zfh-2* null mutants, specifically in pm8 (Fig. 3E). We cannot rule out the possibility that pm8 may have been extruded from the pharynx in *zfh-2* mutants, although this is a less parsimonious scenario.

We also observed tissue gaps between the pharynx and the intestine, indicating that the attachment of these two organs, mediated by the vpi cells, may be defective (Fig. 3B). These cells prominently express ZFH-2 (Fig. 2B) and we therefore examined their proper differentiation and morphology in more detail. We analyzed the expression of the *dmd-4* gene, which encodes a DMRT transcription factor that we previously found to be expressed in four of the six vpi cells, and also in the pm8 pharyngeal muscle (Bayer et al., 2020), which is adjacent to the vpi cells and expresses ZFH-2 as well. Using an endogenously *gfp*-tagged *dmd-4* locus, we found that DMD-4 protein expression in pm8 is partially affected in *zfh-2* null mutants, mirroring the defects of *myo-2* expression. DMD-4 expression in vpi cells was unaffected but the vpi cells appeared highly disorganized (Fig. 3E).

Since *zfh-2* is prominently expressed in the excretory canal cell, as well as in the associated CAN neurons, and since loss of excretory cell function can also result in first larval stage arrest (Nelson and Riddle, 1984), we considered *zfh-2* function in this kidney-like cell type. *zfh-2* null mutant animals do not display the vacuolated, 'clear' (Clr) phenotype that is characteristic of removal of the excretory cell, but we nevertheless examined excretory cell specification in *zfh-2* null mutants using excretory cell reporter transgenes (*exc-4* and *vha-5*). We found that these markers are still expressed and that the overall morphology of the excretory cell appears normal (Fig. S3A,C). An F-actin reporter, LifeAct::TagRFP, expressed specifically in the excretory cell via the promoter of the *glt-3* gene (Shaye and Greenwald, 2015) further corroborates that *zfh-2* does not affect excretory cell fate or morphology (Fig. S3B).

## Analysis of neuronal specification in *zfh-2* null mutants

Given the previously reported function of many homeobox genes in neuronal identity specification (Hobert, 2021), we used a broad panel of terminal identity markers to survey whether complete loss of *zfh-2* results in differentiation defects of the many neurons that express *zfh-2*. Our choice of markers was guided by well-known molecular features of neurons that express the ZFH-2 protein during all larval and adult stages. These markers include *eat-4/VGLUT*, a glutamatergic neurotransmitter identity marker for all the anterior ganglion neurons that express ZFH-2 (IL1, OLQ, URY) (Serrano-Saiz et al., 2013), *unc-17/VAChT*, a cholinergic neurotransmitter identity marker that marks all the cholinergic ventral nerve cord neurons (Pereira et al., 2015), which are all ZFH-2 positive, as well as monoaminergic neuron markers (*cat-1/VMAT, cat-2/TH, tph-1/TPH* and *tdc-1/TDC*) that label several dopaminergic, tyraminergic and serotonergic neuron classes expressing ZFH-2 (CEP, ADE, RIM, NSM, ADF) (Wang et al., 2024). In addition, we analyzed *ins-6*, an insulin marker expressed in ASJ, and a neuropeptidergic marker, *nlp-12*, expressed in DVA, a prominent stretch receptor neuron that also expresses ZFH-2. We also examined proper neuron morphology and differentiation using a dye-filling assay that labels several ZFH-2-expressing neurons (ASH, ASJ, AWB). Since previous mutant analysis had uncovered homeobox regulators for most neurons, except some ZFH-2-expressing neurons (particularly ADF, ASJ, RIM and DVA), we were expecting to discover roles of *zfh-2* in these cells. However, examining all these markers, as well as dye uptake of amphid sensory neurons (including ASJ), first larval stage *zfh-2* null mutant animals showed no obvious defects in expression of these markers or dye uptake (Fig. 4). No obvious defects in overall cellular organization in individual ganglia were observable. One of these markers, *nlp-12*, is cytoplasmically localized in DVA and we observed a normal morphology of the DVA neurite in *zfh-2* null mutants (Fig. 4).

Although we cannot exclude that normally ZFH-2-expressing neurons display some differentiation or functional defects in the absence of *zfh-2*, we can conclude that (1) these neurons are generated and (2) *zfh-2* is unlikely to act, like many other homeobox genes, as a terminal selector-type transcription factor, in which case we would have expected effects on the expression of, for example, neurotransmitter pathway genes, one of the many core identity features regulated by terminal selectors (Hobert, 2016).

The expression of *zfh-2* in all ectodermal worm glia also prompted us to assess their proper generation in *zfh-2* null mutants. We assessed three different glia fate markers, *spig-2* (*txt-17*), *mam-5* and *mir-228*, which label sheath glia, socket glia and all glia, respectively (Aguilar and Hobert, 2024; Fung et al., 2020; Pierce et al., 2008; Miska et al., 2007). We found that the number of cells expressing each of these markers did not significantly differ between wild-type and *zfh-2* null animals, indicating that both sheath and socket glial types are still generated in *zfh-2* null animals (Fig. S1B).

## Post-embryonic removal of ZFH-2 reveals continuous function of ZFH-2 in the alimentary system

To expand our mutant analysis to post-embryonic stages, we sought to circumvent the early larval arrest phenotype of *zfh-2* null mutants by generating a conditional *zfh-2* allele. We used an improved version of the auxin-inducible degron (AID) system (Sepers et al., 2022; Yesbolatova et al., 2020) to remove ZFH-2 protein post-embryonically. With this approach, we expected to not only circumvent early larval arrest, but also to assess potential continuous functions of ZFH-2 in controlling pumping behavior of the embryonically generated pharynx. We used the CRISPR/Cas9 system to insert two mIAA7 tags, flanking a red fluorescent reporter, mScarletI3, at the 3′ end of the *zfh-2* locus (Fig. 2A). Using

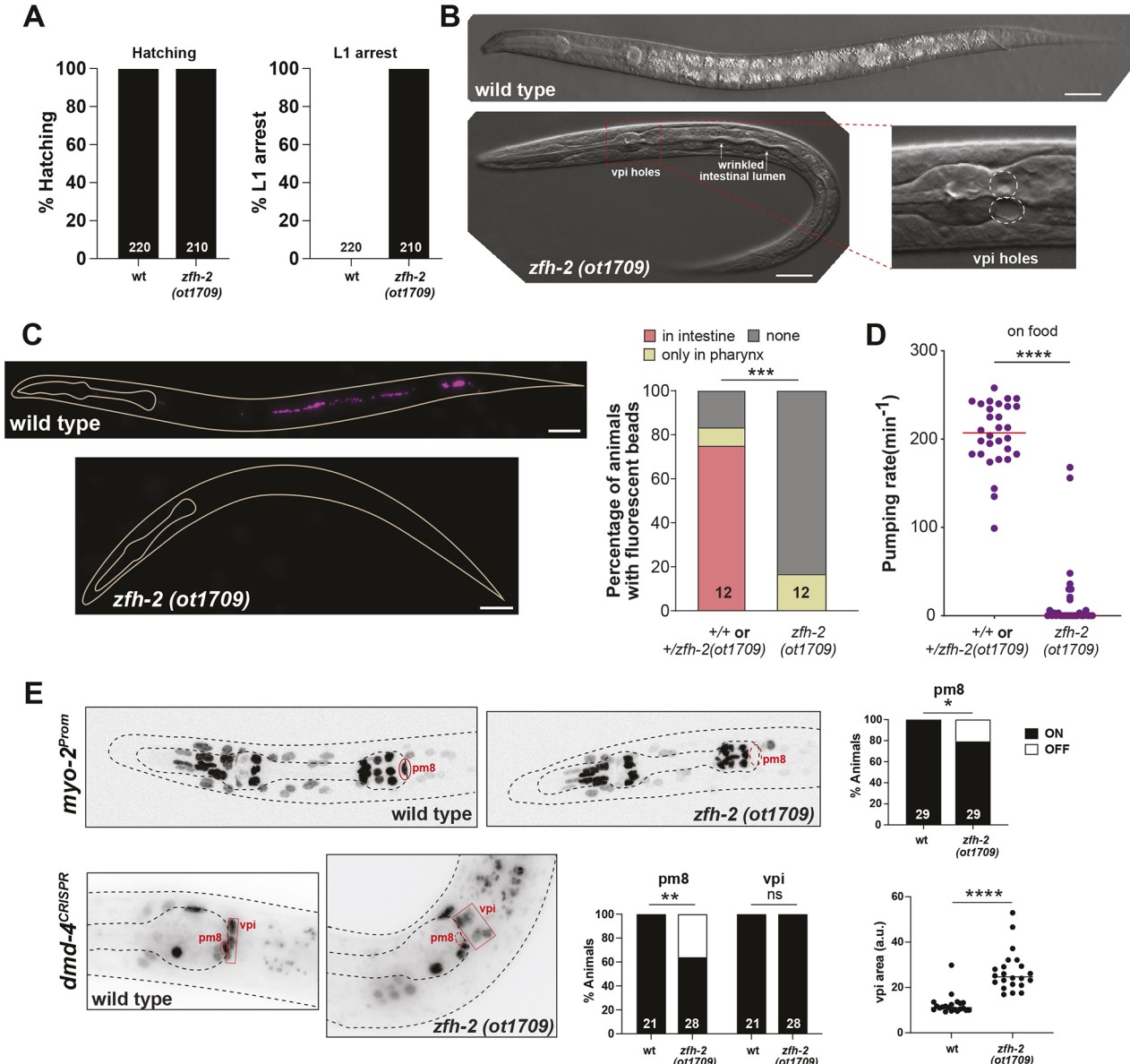

**Fig. 3. *zfh-2* null mutants starve to death at the first larval stage.** (A) Graphs showing the percentage of hatched embryos and arrested L1 larvae in wild type and *zfh-2(ot1709)* null mutants. Sample number is indicated within each bar and represents the number of animals scored. (B) DIC images illustrating holes in the pharyngeal-intestinal valve, thin wrinkled intestine and scrawny nature of *zfh-2(ot1709)* null mutants. Scale bars: 20 μm. (C) Left: Fluorescence images of wild-type and *zfh-2(ot1709)* L1 larval-stage animals fed with red fluorescent beads to measure food intake. The body and pharynx of animals are outlined. Scale bars: 20 μm. Right: Fraction of animals with red fluorescent beads in midgut (intestine), foregut (pharynx) only or in no segments of the gut. ****P*<0.001 (Fisher's exact test). Sample number is indicated within each bar and represents the total number of animals scored. (D) Pharyngeal pumping rate of *zfh-2(ot1709)* L1 larval-stage animals on food. Horizontal line in the middle of data points represents median value of biological replicates. *****P*<0.0001 (Mann–Whitney test). *n*=30. (E) Representative images and quantification showing *myo-2(oxIs322)* and *dmd-4(ot935)* expression in pm8 and vpi. Aside from *myo-2*, the *oxIs322* transgene also contains a *myo-3* reporter that marks body wall muscle cells, which correspond to the nuclei outside the pharynx in wild-type and *zfh-2* mutant images. Dashed lines delineate outline of worms and their pharynges. Animals were scored at the L1 stage. **P*<0.05, ***P*<0.01 (Fisher's exact test). Sample number is indicated within each bar and represents number of animals scored. Bottom right panel shows disorganization of vpi cells in *zfh-2(ot1709)* null mutants. The area of the smallest possible polygon containing all four *dmd-4*-expressing vpi cells was measured as a proxy for vpi organization. *****P*<0.0001 (unpaired *t*-test). a.u., arbitrary units; ns, not significant; wt, wild type.

a ubiquitously expressed, optimized TIR1 protein (TIR1$^{F79G}$) (Negishi et al., 2022), we grew ZFH-2::mScarletI3::mIAA7 worms on auxin plates from the mid-L1 stage onward. We found that the rate of pharyngeal contractions in adult animals with ZFH-2 depleted from mid-L1 was strongly inhibited, with a quarter of the animals not undergoing any foregut contractions (Fig. 5A). Similar to the pumping defect of L1 stage *zfh-2* null mutant (Fig. S2), the strong inhibition of pharyngeal contractions in adults with mid-L1

onward removal of ZFH-2 persisted even after the addition of exogenous serotonin (Fig. 5B), indicating that both the embryonic and post-embryonic roles of *zfh-2* in regulating feeding behavior are serotonin independent.

To test whether ZFH-2 is required to maintain proper feeding behavior in a fully mature animal, we removed ZFH-2::mIAA7 protein by addition of auxin at the mid-L4 stage. Growth of ZFH-2::mIAA7 animals on auxin plates from the mid-L4 stage onward resulted in a

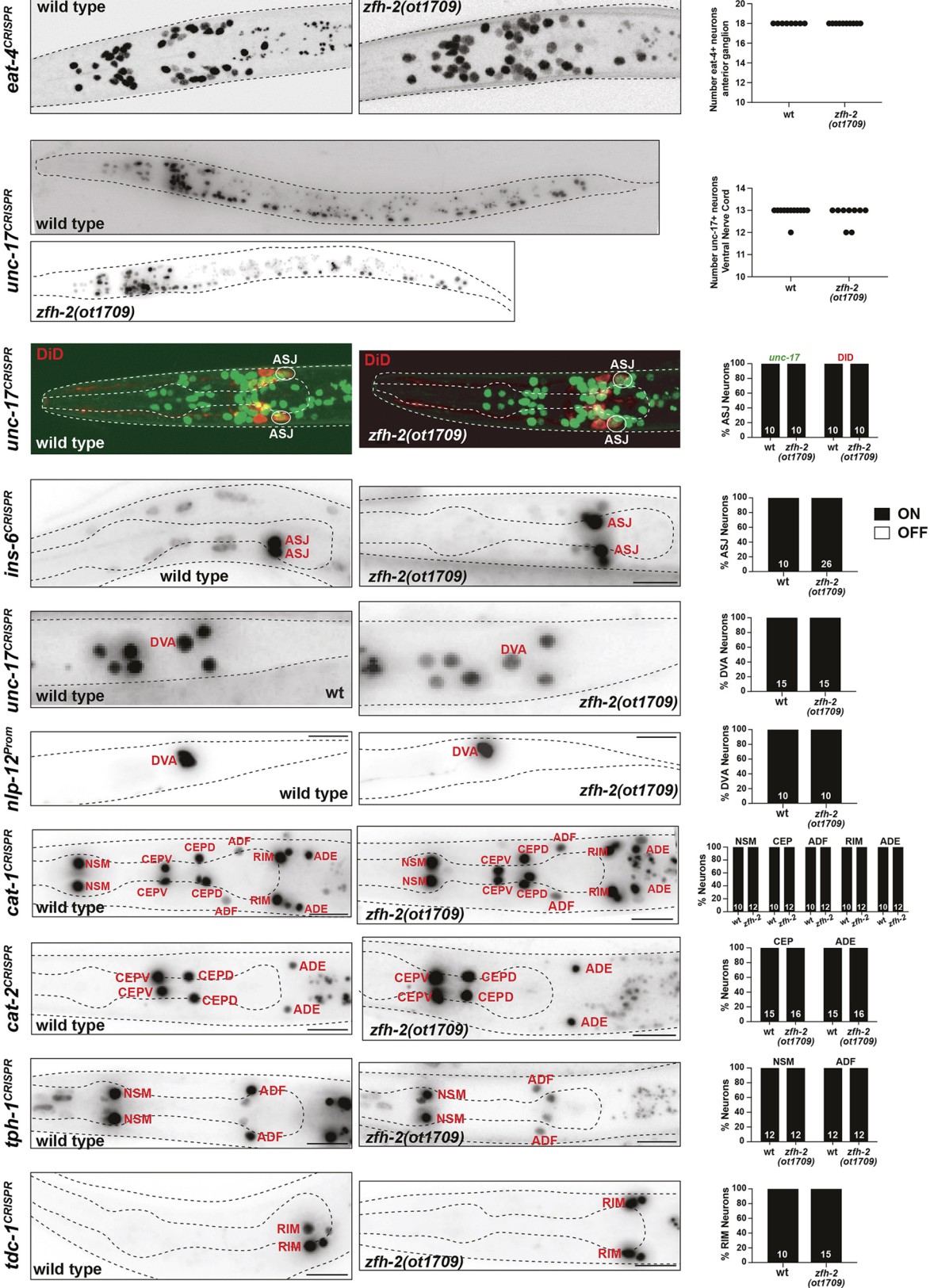

**Fig. 4. zfh-2 null mutants show no obvious defects in nervous system development.** Representative images and quantification showing expression of several neuronal genes in *zfh-2(ot1709)* null mutants. Reporter genes used are CRISPR/Cas9-engineered reporter alleles for *eat-4(syb4257)*, *unc-17(syb4491)*, *ins-6(syb5463)*, *cat-1(syb6486)*, *cat-2(syb8255)*, *tph-1(syb6451)*, *tdc-1(syb7768)* and a transgenic reporter for *nlp-12(otIs706)*. The non-neuronal expression of the *tph-1* reporter is from pharyngeal muscles. Animals were scored at the L1 stage. Sample number is indicated within each bar and represents the number of animals scored. Dashed lines delineate outline of worms and their pharynges. DiD, commercial lipophilic dye (Invitrogen, V22887); wt, wild type. Scale bars: 10 μm.

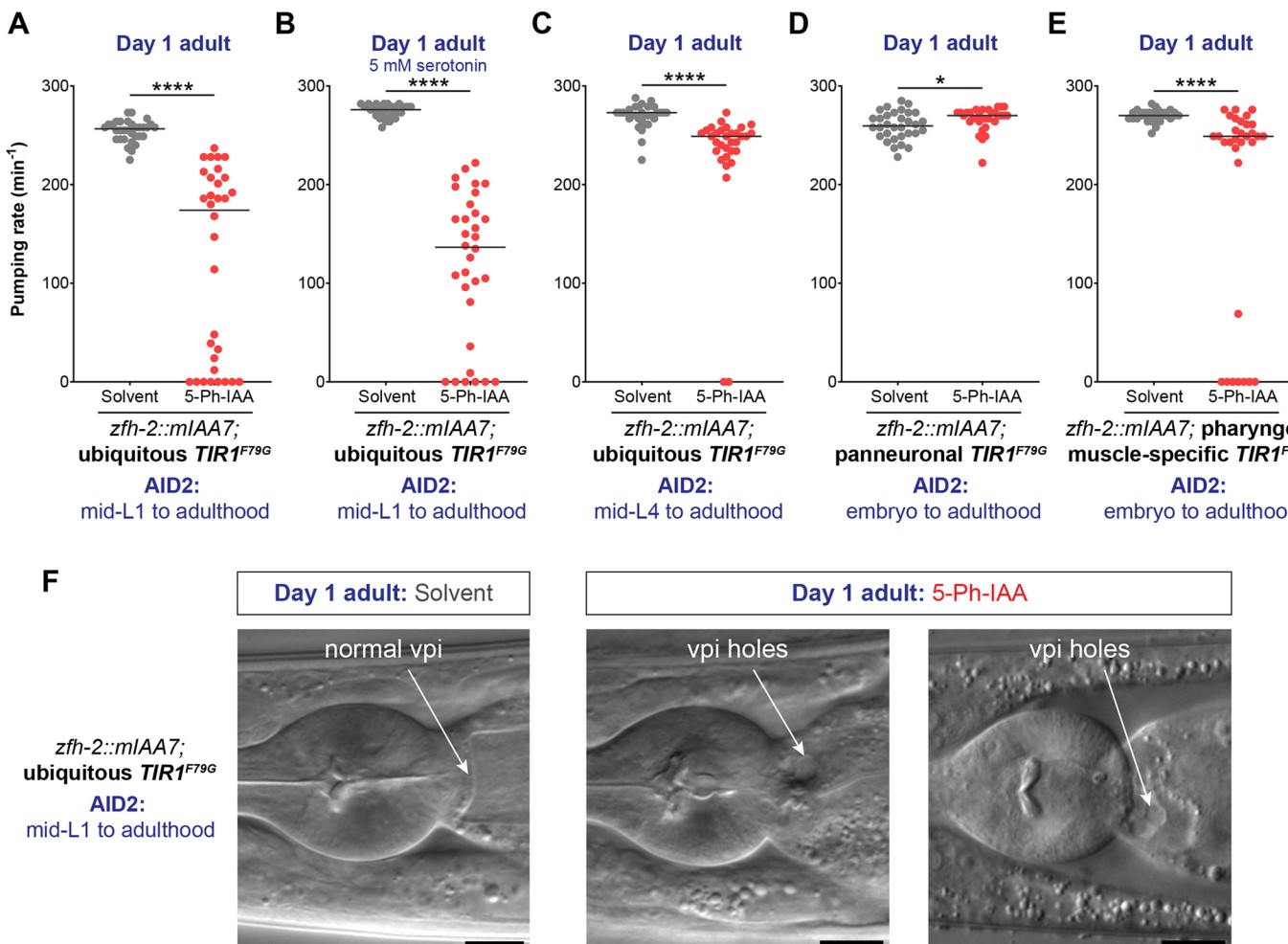

**Fig. 5. Post-embryonic removal of ZFH-2 protein results in pumping defects.** (A) Pharyngeal pumping rate of *zfh-2(syb10278); osIs158[eft-3p::TIR1(F79G)]* adults subjected to ubiquitous ZFH-2 depletion starting at mid-L1 stage. Animals were treated with either solvent (ethanol) or 100 µM 5-Ph-IAA starting at mid-L1. (B) Pharyngeal pumping rate of *zfh-2(syb10278); osIs158[eft-3p::TIR1(F79G)]* adults in the presence of 5 mM serotonin after ubiquitous ZFH-2 depletion starting at mid-L1 stage. Animals were treated with either solvent (ethanol) or 100 µM 5-Ph-IAA starting at mid-L1. (C) Pharyngeal pumping rate of *zfh-2(syb10278); osIs158[eft-3p::TIR1(F79G)]* adults subjected to ubiquitous ZFH-2 depletion starting at mid-L4 stage. Animals were treated with either solvent (ethanol) or 100 µM 5-Ph-IAA starting at mid-L4. (D) Pharyngeal pumping rate of *zfh-2(syb10278); otIs935[UPNp:TIR1(F79G)]* adults subjected to pan-neuronal ZFH-2 depletion starting at embryo stage. Animals were treated with either solvent (ethanol) or 100 µM 5-Ph-IAA throughout all developmental stages. (E) Pharyngeal pumping rate of *zfh-2(syb10278); otSi4[myo-2p::TIR1(F79G)]* adults subjected to ZFH-2 depletion only in pharyngeal muscles starting at embryo stage. Animals were treated with either solvent (ethanol) or 100 µM 5-Ph-IAA throughout all developmental stages. In A-E, horizontal line in the middle of data points represents median value of biological replicates. *$P<0.05$, ****$P<0.0001$ (Mann–Whitney test). $n=32$. (F) DIC images of the pharyngeal-intestinal valve (vpi) in *zfh-2(syb10278); osIs158[eft-3p::TIR1(F79G)]* adults subjected to ubiquitous ZFH-2 depletion starting at mid-L1 stage. Animals were treated with either solvent (ethanol) or 100 µM 5-Ph-IAA starting at mid-L1. Scale bars: 10 µm.

mild but significant reduction in their rate of pharyngeal pumping (Fig. 5C). We conclude that ZFH-2 is continuously required after embryonic formation of the alimentary tract to promote food ingestion.

### ZFH-2 acts in foregut muscles, not neurons, to control pharyngeal pumping

The ZFH-2::mIAA7 allele also enabled us to analyze the site of action of ZFH-2 in regard to the pharyngeal pumping defects. To this end, we generated transgenic lines in which TIR1^F79G is either expressed in all pharyngeal muscles, using the *myo-2* promoter or in all neurons, using the UPN promoter (Yemini et al., 2021). Depletion of ZFH-2 with the pan-neuronal TIR1 strain did not result in a reduction of pharyngeal pumping behavior, while depletion of ZFH-2 from muscle resulted in pumping effects (Fig. 5D,E). Together with the effects of ZFH-2 on *myo-2* expression in pm8 (Fig. 3E), we infer that ZFH-2 functions in pharyngeal muscle to

control feeding behavior. We note that these pumping defects are not as strong as what we observed upon ubiquitous ZFH-2 depletion, indicating additional cellular site(s) of action. One such site may be the vpi since in adult animals with ubiquitous ZFH-2 depletion from mid-L1, we observed holes in the vpi region (Fig. 5F). We ascribe this to a continuous requirement of ZFH-2 in maintaining the structural integrity of the vpi, which may affect pumping behavior as well.

### Effect of post-embryonic removal of ZFH-2 on nervous system function

While we did not observe any overt neuronal development defects in *zfh-2* null mutants, we tested whether post-embryonic removal of ZFH-2 protein results in obvious neuron function defects in the adult animal (the larval arrest of null mutants prevented behavioral analysis of adult null mutants). We were motivated to undertake this analysis by two observations: (1) *zfh-2* is expressed in the main

nociceptive neuron of *C. elegans*, ASH (Kaplan and Horvitz, 1993), as well as several of its synaptic command interneuron targets which mediate nociceptive responses perceived by ASH; (2) human *ZFHX2* variants are associated with a pain insensitivity phenotype, Marsili syndrome (Habib et al., 2018). We used the AID allele to remove ZFH-2 from all ZFH-2-expressing cells after the L1 stage and assayed adult animals for their avoidance response to a noxious, ASH-sensed cue, octanol. We found that post-embryonic ZFH-2 removal does not affect the ability of animals to sense octanol, or to initiate a locomotory reversal response (Fig. S4). Notably, these animals respond to octanol even more quickly than control animals (Fig. S4), perhaps as a secondary consequence of the feeding defects of these animals which may result in greater sensory acuity. The avoidance response not only indicates that the perception of noxious stimuli is intact but also indicates that *zfh-2* is not required for proper locomotory behavior. Hence, despite the prominent expression of ZFH-2 in all cholinergic motor neurons and innervating command interneurons, ZFH-2 depleted animals show none of the obvious locomotory defects associated with defects in the function of these neurons.

### Post-embryonic removal reveals function of ZFH-2 in spermatheca and vulval development

Apart from feeding defects, we noted that ubiquitous, post-embryonic removal of ZFH-2::mIAA7 produced striking defects in several post-embryonically generated structures of the gonad and egg laying apparatus. First, the spermatheca, a bag-like compartment of the gonad where sperm fertilize oocytes, normally clearly visible by light microscopy, did not appear to form (Fig. 6A). A few sperm localized around the expected location of the spermatheca (Fig. 6A). The lack of a recognizable spermatheca is consistent with the prominent expression of ZFH-2 in the spermatheca cells, as described above (Fig. 2B). The overall structure of the female gonad appeared initially fine, but as oogenesis continues, oocytes were pushed further back into the gonad on day 2 of adulthood, due to an apparent inability to be fertilized by sperm (Fig. 6B). A few oocytes appeared to become fertilized, but these fertilized embryos arrested very early in their development and were not laid (Fig. 6). Post-embryonic ZFH-2 removal therefore results in a completely penetrant infertility of these animals. Consistent with this, we observed yolk accumulation in the pseudocoelom of ZFH-2 depleted animals on day 2 of adulthood (Fig. 6B), which is generally a hallmark of post-reproductive adults (Ezcurra et al., 2018; Herndon et al., 2002).

We also observed a protruding vulva in post-embryonically ZFH-2-depleted animals, often resulting in animals with vulva ruptures ('exploding vulva') (Fig. 6). This phenotype may relate to expression of ZFH-2 in uterine and vulval cells (Fig. 2).

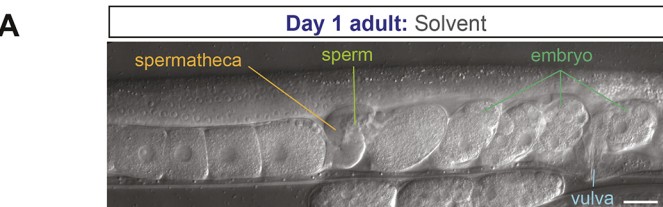

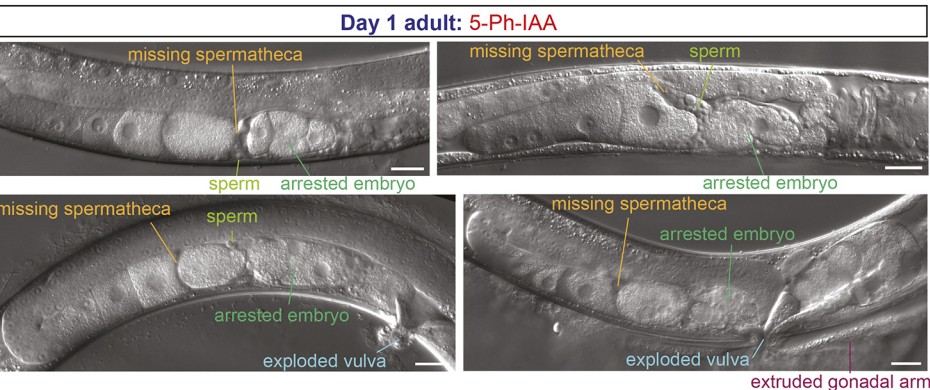

**Fig. 6. Post-embryonic removal of ZFH-2 protein results in spermathecal and vulval defects.** (A,B) DIC images of the gonad of *zfh-2(syb10278); osIs158[eft-3p::TIR1(F79G)]* day 1 (A) and day 2 (B) adults subjected to ubiquitous ZFH-2 depletion starting at mid-L1 stage. Animals were treated with either solvent (ethanol) or 100 µM 5-Ph-IAA starting at mid-L1. Scale bars: 20 µm.

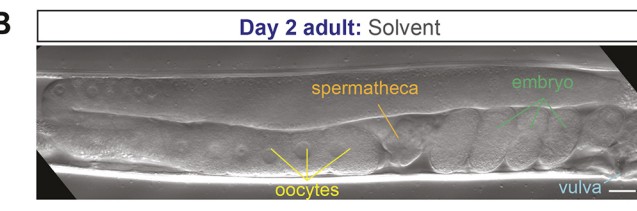

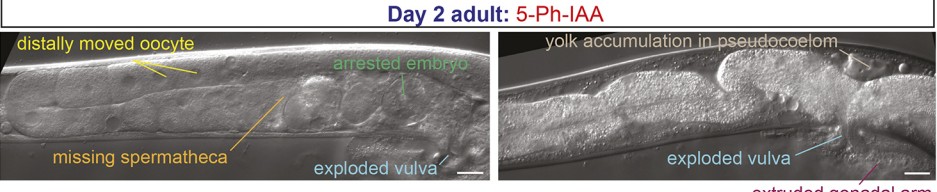

## Only a subset of the *zfh-2* isoforms provides essential gene function

Armed with the knowledge of the *zfh-2* null phenotype, as well as phenotypes observed upon conditional removal of ZFH-2 protein, we set out to assess the function of individual isoforms of the *zfh-2* locus and examined the contribution of individual zinc fingers and homeodomains for ZFH-2 protein function. One motivation behind such molecular dissection was the question of whether the function of *zfh-2* in distinct cell types can be genetically separated such that individual isoforms or domains are allocated to function in specific cellular contexts.

To analyze the individual isoforms (shown in Fig. 7A), we first introduced premature stop codons into exons specific for the longest, first isoform of the locus, *zfh-2a*. We found that these animals grow to become fertile adults, display no obvious morphological defects and exhibit only a very mild defect in pharyngeal pumping (Fig. 7). Similarly, animals carrying a deletion allele, *tm12720*, that internally deletes the first exons of the second and third isoform, *zfh-2b* and *zfh-2c*, respectively, were also homozygous viable with no obvious defects in gonad structure, fertility or pumping (Fig. 7). In contrast, the introduction of a nonsense codon, *ot1809*, into the tenth exon of the locus, which also affects the production of the fourth isoform, *zfh-2d*, resulted in an almost completely penetrant larval arrest phenotype and strongly reduced pharyngeal pumping (Fig. 7). The few animals that made it to adulthood had a scrawny, starved appearance and were completely sterile (Fig. 7B). The phenotype is

not quite as strong as the null phenotype since larval arrest often happens after the first larval stage.

A small deletion that affects all isoforms except the smallest isoform (*tm310* allele; Fig. 7A), had a more severe phenotype than the nonsense mutation in the *zfh-2a-d* isoforms. These animals showed a completely penetrant larval arrest phenotype and a very strong pumping defect, albeit still not as severe as the null allele (Fig. 7B,C). We conclude that the shorter isoforms of the *zfh-2* locus, all containing the complete set of homeoboxes, are key to providing the essential function of the locus that we described here.

### Importance of the homeoboxes for *zfh-2* gene function

Our mutational analysis of isoform requirements indicates that the first six zinc fingers are mostly dispensable for ZFH-2 protein function, at least within the contexts described here, but they do not address the functional contributions of the homeodomains of the ZFH-2 protein. We focused our analysis on homeodomain 2 and homeodomain 3, because they are more conserved to the homeodomains of other ZFHX proteins than homeodomain 1 is (Fig. 1).

Homeobox 2 and 3 are each completely contained within an exon and we deleted each domain either alone or together using the CRISPR/Cas9 system (Fig. 8A). Animals in which we deleted the third homeodomain alone (*zfh-2^{ΔHD3}*) were homozygous viable, morphologically normal, produced progeny and did not display pumping defects (Fig. 8B,C).

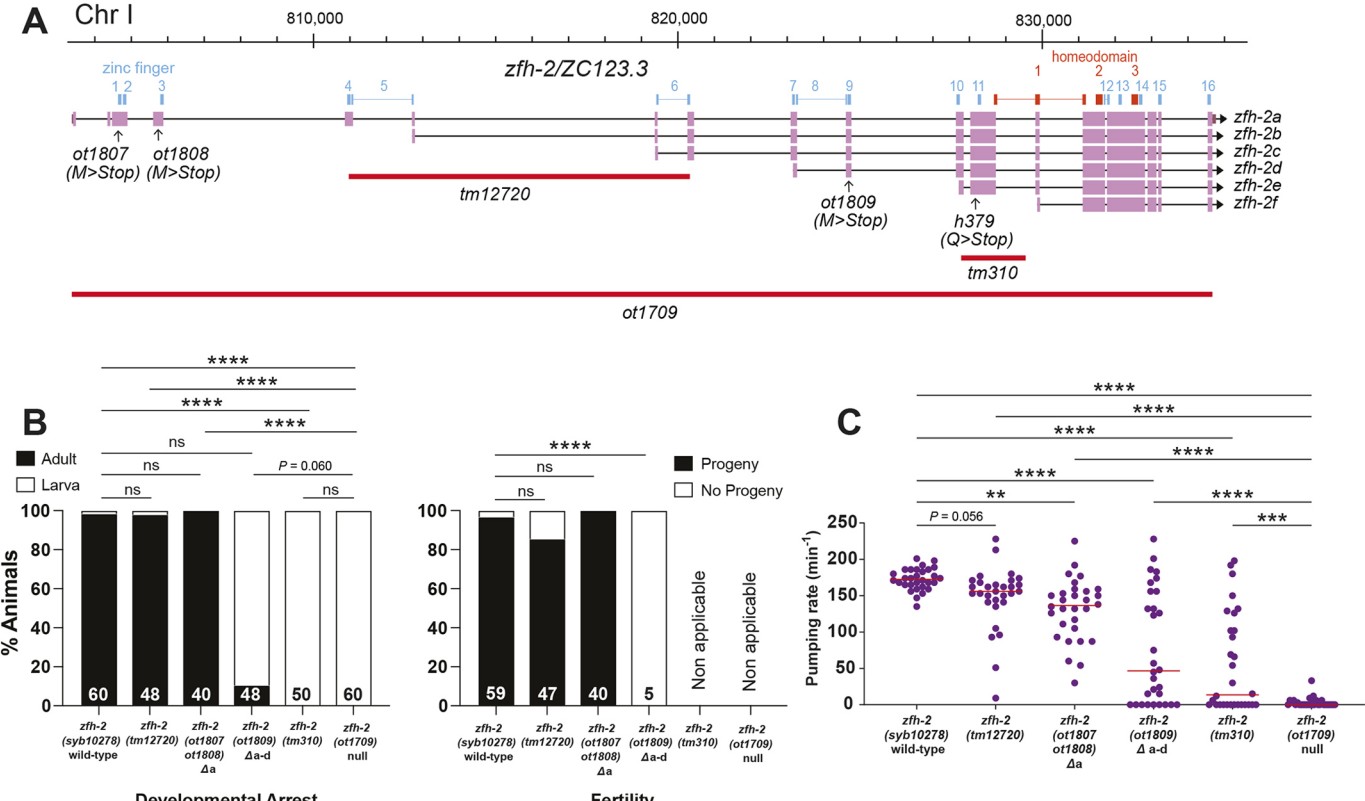

**Fig. 7. Only a subset of the *zfh-2* isoforms provides essential gene function.** (A) Schematic of the *zfh-2* locus showing the isoform-specific alleles analyzed. (B) Effect of isoform-specific mutants on development and fertility. Graphs showing percentage of animals arresting during larval development (left) or giving progeny (right). ****$P<0.0001$ (Holm–Šídák's multiple comparisons test after Fisher's exact test). Sample number is indicated within each bar and represents the number of animals scored. (C) Pharyngeal pumping rate of isoform-specific mutants and existing *tm* alleles of *zfh-2* at L1 larval stage on food. Horizontal line in the middle of data points represents median value of biological replicates. **$P<0.01$, ***$P<0.001$, ****$P<0.0001$ (Holm–Šídák's multiple comparisons test after one-way ANOVA). ns, not significant.

In contrast, animals carrying a deletion of the second homeodomain ($zfh$-$2^{\Delta HD2}$) could not be maintained as homozygous mutant animals. Although the vast majority of these animals grew up to become adults, they had a scrawny appearance and were completely sterile (Fig. 8B), with spermathecal and vulva defects that were indistinguishable from those observed after post-embryonic depletion of ZFH-2 (Fig. 8D).

The strong pumping defect of these animals, already apparent at the first larval stage, is likely responsible for their scrawny appearance (Fig. 8C).

Deletion of the third homeodomain as well as the second homeodomain deletion ($zfh$-$2^{\Delta HD2\Delta HD3}$) worsened the severity of these defects (Fig. 8B,C). Consistent with even stronger pumping

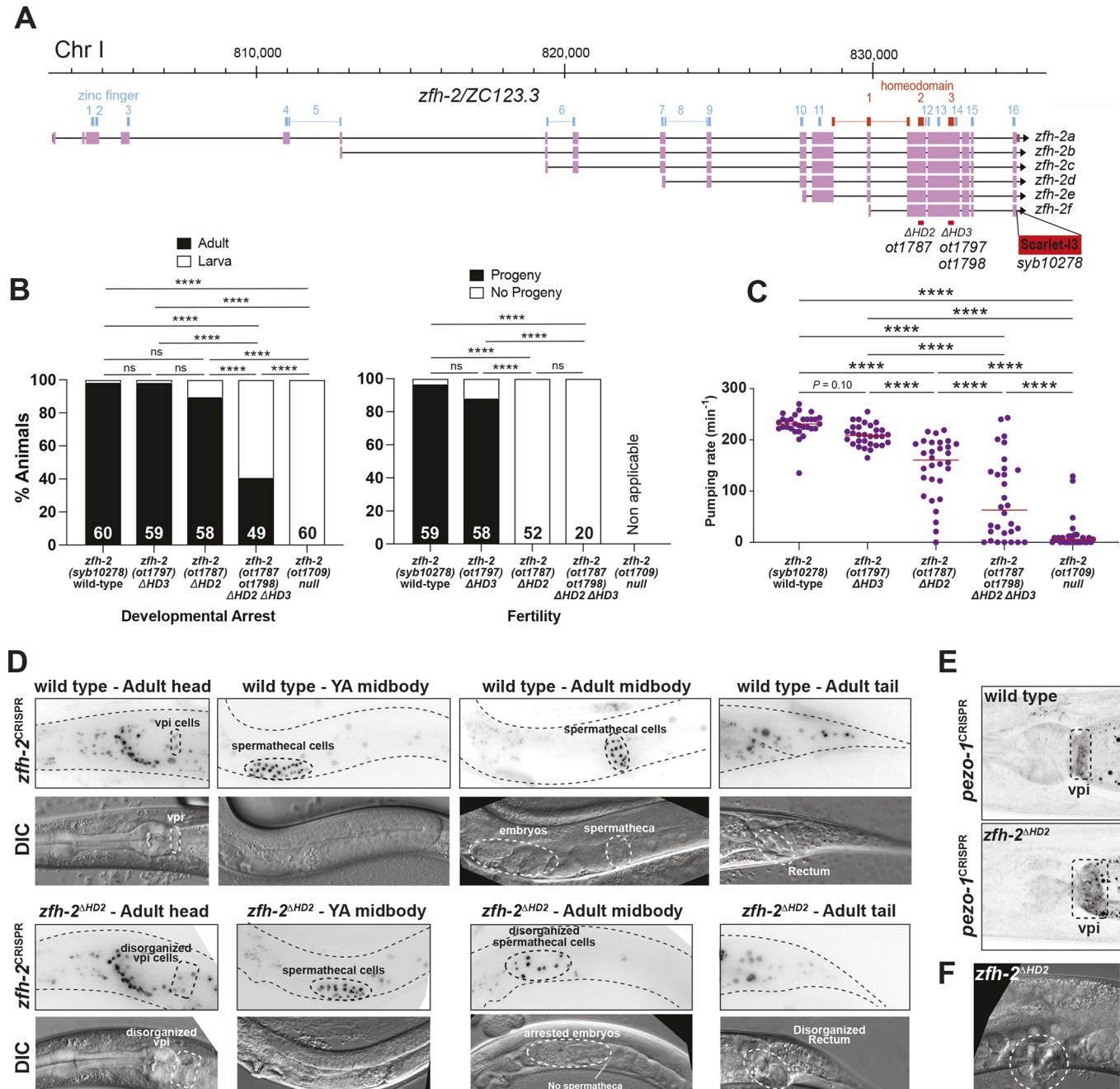

**Fig. 8. Homeoboxes are required for *zfh-2* gene function.** (A) Schematic of the *zfh-2* locus showing the homeodomain deletion alleles analyzed. (B) Effect of homeodomain deletion mutants on development and fertility. Graphs showing percentage of animals arresting during larval development (left) or giving progeny (right). ****$P$<0.0001 (Holm–Šídák's multiple comparisons test after Fisher's exact test). Sample number is indicated within each bar and represents the number of animals scored. (C) Pharyngeal pumping rate of homeodomain deletion alleles of *zfh-2* at L1 larval stage on food. Horizontal line in the middle of data points represents median value of biological replicates. ****$P$<0.0001 (Holm–Šídák's multiple comparisons test after one-way ANOVA). ns, not significant. (D) Visualizing disorganization of the vpi and spermatheca with the *zfh-2(syb10278)* CRISPR reporter allele in which the second homeobox was deleted [*zfh-2$^{\Delta HD2}$(ot1787)*]. DIC images of *zfh-2$^{\Delta HD2}$(ot1787)* mutants showing disorganized vpi, spermatheca and rectum. Dashed lines delineate outline of worms. YA, young adult. (E) Disorganization of the vpi in *zfh-2$^{\Delta HD2}$(ot1787)* mutants with a *pezo-1(av146)* reporter allele (Bai et al., 2020). (F) DIC image of protruding vulva in *zfh-2$^{\Delta HD2}$(ot1787)* mutants.

defects (Fig. 8C), more than half of these animals arrested at larval stages and those that reached adulthood were scrawny and completely sterile (Fig. 8B). This phenotype is still not as severe as the 100% penetrant first larval arrest phenotype of *zfh-2* null mutant animals (Fig. 8B).

## The *zfh-2*$^{\Delta HD2}$ allele corroborates additional functions of the *zfh-2* locus

Homeodomain deletions do not simply destabilize the ZFH-2 protein. Since we conducted the homeodomain deletion in the context of a fluorophore tagged *zfh-2* locus (Fig. 8A), we were able to confirm that ZFH-2$^{\Delta HD2}$::mScarletI3 is still expressed in these animals (Fig. 8D). The ability to visualize cells with apparently dysfunctional ZFH-2 protein confirmed the impact of ZFH-2 on the vpi cells, as well as on the spermatheca. Specifically, in the gonad, the ZFH-2$^{\Delta HD2}$:: mScarletI3 signals revealed that the spermathecal cells are in fact present, and in young adult worms form a somewhat recognizable structure, but as oocytes start pushing through, the cells become highly disorganized, and the bag-like structure disintegrates (Fig. 8D). The vpi cells, visualized again with ZFH-2$^{\Delta HD2}$::mScarletI3, appeared to be mispositioned and there were apparent holes in the vpi area in *zfh-2*$^{\Delta HD2}$ mutant animals (Fig. 8D).

We also used the *zfh-2*$^{\Delta HD2}$ mutant allele to test another molecular feature of the vpi cells, namely the expression and localization of the mechanosensory PIEZO channel PEZO-1. This channel has been reported to localize to the membrane of pharyngeal intestinal valve cells and to be required for food swallowing (Bai et al., 2020; Park et al., 2024). A tagged *pezo-1* reporter allele, crossed into a *zfh-2*$^{\Delta HD2}$ mutant background showed that *pezo-1* gene expression is unaffected, but corroborated the requirement of *zfh-2* for the structural integrity of the vpi cells, which appeared enlarged and disorganized (Fig. 8E).

Similar to what we observed with conditional, post-embryonic ZFH-2 protein depletion, *zfh-2*$^{\Delta HD2}$ mutant animals also displayed a protruding vulva phenotype (Fig. 8F).

Lastly, growth of *zfh-2*$^{\Delta HD2}$ mutant animals to the adult stage also allowed us to notice defects around the rectal canal that is formed by strongly ZFH-2-expressing epithelial cells. We observed empty spaces that may be indicative of incorrect formation of this tubular structure (Fig. 8D). Together with our analysis of distinct isoforms of the locus, we conclude that the homeodomains are required for all presently identified functions of ZFH-2.

## DISCUSSION
Each neuron class in the *C. elegans* nervous system expresses a unique combination of homeodomain transcription factors. For 113 of the 118 neuron classes at least one homeobox gene has been implicated in their differentiation; four of the five remaining neuron classes express the *zfh-2* zinc finger homeobox gene. Hence, this study was aimed at investigating roles for *zfh-2* in these neurons, or any of the other neuron classes in which *zhf-2* is expressed. However, using a panel of key neuronal identity markers, we were unable to detect any defects in the differentiation of these neurons upon complete elimination of the *zfh-2* locus. *zfh-2* null mutants also do not show behavioral defects characteristic of the loss of function of many of the normally *zfh-2*-expressing neurons. For example, the locomotory behavior of *zfh-2* null mutant appears intact even though *zfh-2* is normally expressed in all cholinergic motor neurons of the ventral nerve cord and their innervating command interneurons. We consider it also quite unlikely that loss of *zfh-2* can be compensated for by the other *C. elegans* zinc finger homeobox gene *zag-1*, since their expression shows only limited overlap (Reilly et al., 2020). We conclude that ZFH-2 has none of

the terminal selector functions associated with many other homeodomain proteins (Hobert, 2021).

While we were preparing this manuscript for publication, an ahead-of-print study reported that ZFH-2 acts in physical association with DAF-16 to control the lifespan extension observed upon elimination of insulin-like signaling in *C. elegans* (Artan et al., 2025). Previous work had shown that DAF-16 acts both in the nervous system (where DAF-16 expression overlaps with ZFH-2) and the intestine (where there is minimal overlap of ZFH-2 and DAF-16) to control lifespan (Uno et al., 2021; Zhang et al., 2022). Together with the lack of apparent developmental defects in the nervous system, we infer that neurons require ZFH-2, not for their differentiation, but for functional features that enable them to control lifespan, perhaps through the regulation of signals emanating from the nervous system.

We observed prominent functions of *zfh-2* in non-neuronal cell types. Congruent with *zfh-2* expression in pharyngeal muscle and valve cells that link the pharynx (foregut) with the intestine (midgut), we found that the pharynx of *zfh-2* mutants is unable to pump and ingest food. This is, at least in part, due to the pm8 differentiation defects and vpi disorganization in *zfh-2* null mutants, as shown with the *myo-2*, *dmd-4* and *pezo-1* markers. Since conditional removal of ZFH-2 after the development of all pharyngeal tissue results in strong pumping and ingestion defects, we can conclude that *zfh-2* is required to maintain the functional state of some of these cells. A previous study described a role of an embryonic Notch signal in pm8 myogenesis and morphogenesis, including pm8 defects in *myo-2* expression in Notch mutants (Rasmussen et al., 2008). While our conditional *zfh-2* removal experiments argue for *zfh-2* functioning at least in part after receipt of the embryonic Notch signal, it is conceivable that *zfh-2* acts to properly interpret and manifest this embryonic Notch signal.

The other main tissue type in which we found ZFH-2 to act is the gonad, where ZFH-2 affects the structural integrity of the spermatheca, a bag-like chamber made of a group of epithelial cells that houses sperm and is the site of oocyte fertilization. Although we cannot infer the null phenotype of ZFH-2 in the spermatheca (due to the earlier larval arrest phenotype), the homeodomain 2 deletion of the ZFH-2 protein shows that spermathecal cells are generated and originally form a pseudo-organized chamber in young adult worms that later loses its integrity and becomes profoundly disorganized, resulting in complete sterility of the animal. Sterility caused by mutations that affect specific aspects of germline development have been reported to result in increased fat accumulation in the intestine (Chaturbedi and Lee, 2025) and, hence, the absence of spermatheca and resulting infertility may be the reason why previous incomplete *zfh-2* depletion by RNAi resulted in increased intestinal fat staining (Ke et al., 2021).

A common theme in ZFH-2 function appears to be that it does not affect the overall lineage, generation or fate of the cells it acts in, but rather affects their proper assembly into functional units. Particularly in the context of the vpi, the spermatheca, vulva and the rectal canal, it is attractive to speculate that ZFH-2 may regulate the expression of cell surface molecules required for the proper arrangement of cells into tubular structures. Similarly, the function of ZFH-2 in the nervous system may lie in properly organizing or re-organizing cell–cell contacts under specific conditions and/or maintain them during aging.

The unusual accumulation of DNA-binding domains (zinc fingers and homeodomains) in a single transcription factor raises the question of which of these many domains is required for function. We have undertaken here, to our knowledge, the first structure/function analysis of ZFHX transcription factors and demonstrate the

importance of its homeodomains within distinct cellular contexts, and the apparent superfluousness of many of its zinc finger domains. This holds for at least three distinct sites of action of ZFH-2: the alimentary apparatus, the gonad and the vulva. Future analysis may reveal that ZFH-2 exerts different, as-yet-unrecognized functions in other cell types and these possible additional functions may have different domain requirements.

## MATERIALS AND METHODS
### C. elegans strains
Worms were grown at 20°C on nematode growth media (NGM) plates seeded with *Escherichia coli* (OP50) bacteria as a food source. The wild-type strain used was Bristol N2. A complete list of strains used in this study can be found in Table S2.

### Generation of zfh-2 alleles
*zfh-2(ot1812)* reporter allele was generated by germline Cre recombinase-mediated removal of the *unc-119(+)* rescuing cassette from the *zfh-2(st12167)* original allele. The *unc-119(tm4063)* mutation in the background was outcrossed.

Different mutant alleles for *zfh-2* were generated by CRISPR/Cas9 genome engineering as described (Dokshin et al., 2018) with the following crRNAs and ssODN sequences.

### zfh-2(ot1709) – deletion of full locus
The following crRNAs and ssODN sequences were used: crRNA1: CTACCCATTTAGCCAATATAT; crRNA2: GTAGTAGTAGTAGTATG-AGG; ssODN: AAATTCATCCAAAAAAATTTCCAGAGTTGCCCCGC-CCATACATACTACTACTACTACCACGACGACGCCATAACAAAACC. The resulting strain was L1 lethal and balanced with the aneuploidy-free balancer *tmC20* (Dejima et al., 2018).

### zfh-2$^{\Delta HD2}$(ot1787) – in frame deletion of homeodomain 2
The following crRNAs and ssODN sequences were used: crRNA1: TCCTGCAACACGTCGTCCAG; crRNA2: CGGCGTTCTCACAAATC-GAT; ssODN: ATGACACCGAGCACTCCTTCCTGCAACACGTCGT-CCTcTGGACGAATCTATGAGAATCAGCCGAATCACGAGAGTTCtGA-TCGATTTGTGAGAACGCCGGGATCGAACTTTCAGTGC. This allele was generated in the background of the *zfh-2* reporter allele *syb10278*. The resulting strain was sterile and balanced with the aneuploidy-free balancer *tmC20* (Dejima et al., 2018).

### zfh-2$^{\Delta HD3}$(ot1797) – in frame deletion of homeodomain 3
The following crRNAs and ssODN sequences were used: crRNA1: GCATCGGTGTGAGATGAGTT; crRNA2: GCCAAAGAGCGAAAG-ACGCG; ssODN: CAAGCGGCTCGGAATGCAGATCTCCGGCGAGC-AACACGCGCGGTGCAGTTGACGAGGACTCTCGATCCGGAG. This allele was generated in the background of the *zfh-2* reporter allele *syb10278*. *zfh-2$^{\Delta HD3}$(ot1798)* was generated with the same reagents as *zfh-2$^{\Delta HD3}$(ot1797)* and both alleles are molecularly identical. The difference is that *zfh-2$^{\Delta HD3}$(ot1798)* was generated in the background of *zfh-2$^{\Delta HD2}$(ot1787)* because these two modifications are in the same locus and linked.

### zfh-2$^{\Delta a}$(ot1807 ot1808)
Mutation of isoform a Met133, Met233 and Met234 to STOP codons was used to generate a specific mutant. The following crRNAs and ssODN sequences were used. To generate *ot1807* (Met133 to STOP) – crRNA: GAAGGAGAAGCACAACGATG; ssODN: CCCTTGAAATTCATATG-AAGGAGAAGCACAACGATGTCGATGTCAAGTGCTAGTTTTGTGC-CGAGAACCGTCCCCACCCGAAGCTGGC. To generate *ot1808* (Met233, Met234 to STOP) – crRNA: TTCCACGTGTTCCATCATTT; ssODN: TCT-GCGGTATCTTCGCCACCGAATCTATCGCCGAATAGTAGGAACAC-GTGGAACAAGACCGTTCCAGGACGTTCCA. Everything was done in a single injection. This allele was generated in the background of the *zfh-2* reporter allele *syb10278*.

### zfh-2$^{\Delta d}$(ot1809)
Mutation of isoform d Met50 to STOP codon was used to generate an isoform d mutant. This modification also affects isoforms a, b and c. The following crRNAs and ssODN sequences were used: crRNA: GCACA-TTTCGCATTCTTCTT; ssODN: TCGTTTTTTTTTCAGACCCATATGT-TGGAGCACACgAAAGAAGAATGCGAATAGTGCTCTGAAACATTTG-CCACAAAAGAGGCATTCC. This allele was generated in the background of the *zfh-2* reporter allele *syb10278*. The resulting strain was larval lethal and balanced with the aneuploidy-free balancer *tmC20* (Dejima et al., 2018).

### zfh-2(syb10278)
An AID allele was used for conditional ZFH-2 protein degradation. The *zfh-2* locus was tagged at the C terminus with the sequence GSGGGSGGTGGSG::mIAA7::wrmScarlet-I3::mIAA7. This strain was generated by SunyBiotech.

### Conditional ZFH-2 protein degradation
To generate the pharyngeal muscle-specific *TIR1(F79G)* transgene *otSi4*, an *F79G* [Phe(TTC) to Gly(GGA)] mutation was introduced in the *TIR1* sequence of the single-copy transgene *ieSi60[myo-2p::TIR1::mRuby::unc-54 3′UTR]* using CRISPR/Cas9 genome engineering based on a previously described strategy (Hills-Muckey et al., 2022). The following crRNAs and ssODN sequences were used: crRNA: TTCCCTTGAGCTCGACGGAA; ssODN: CCTCCCCATCCGTCTGGGACGAGGTTGAAGTCGGCTCC-GTGTGGCTTTCCCTTCAATTCGACACTACGGACCTTTGGGAAAC-GACGGATGACGGTGGCTG.

The pan-neuronal *TIR1(F79G)* transgene *otIs935*, containing *UPNp::TIR1(F79G)::mTur2::tbb-2 3′UTR*, is superior to other previously described pan-neuronal TIR1 constructs and will be described elsewhere (G. Valperga and O.H., unpublished).

Conditional ZFH-2 protein degradation using AID2 was performed on NGM plates containing 100 μM 5-phenyl-indole-3-acetic acid (5-Ph-IAA) that were prepared using a previously described protocol (Sural et al., 2024). Animals were transferred to 5-Ph-IAA plates at the desired developmental stage after washing in M9 buffer. Control plates had the same volume of solvent alone (ethanol). All plates were stored in the dark for the entire duration of the experiment.

### Pumping and feeding assays
To visualize intake of food, animals were transferred to NGM plates seeded with a 1:100 mixture of 0.5 μm sized red fluorescent beads (Millipore Sigma, L3280) in concentrated OP50 solution. Animals were allowed to crawl on the plates for 10 min, after which they were imaged using a Zeiss compound microscope (Imager Z2).

Measurement of pharyngeal pumping on food was recorded from animals freely moving on NGM plates seeded with a uniform thin layer of OP50 bacteria. Animals were allowed to settle down for 5 min and the movement of the grinder of the pharynx was recorded using a hand-held tally counter by observing the animals under a Nikon Eclipse E400 upright microscope equipped with DIC optics. For L1 and adult stage animals, the number of grinder movements in a 20 s period was recorded using 50× and 20× air objective lenses, respectively, and the number of recorded pumps was multiplied by three to obtain pharyngeal pumps per minute. The rate of pharyngeal pumping was recorded from at least 15 animals per day and on at least two independent days.

For recording pharyngeal pumping in the presence of serotonin, NGM plates were prepared with a final concentration of 5 mM serotonin hydrochloride (Millipore Sigma, H9523). The plates were dried overnight at room temperature in the dark. Subsequently, animals were transferred to the serotonin plates and were allowed to settle down for 5 min before recording of pharyngeal pumping using the protocol described above.

### Octanol avoidance assay
Octanol assay to assess ASH function was performed as previously described with some modifications (Srinivasan et al., 2008; Troemel et al., 1997). Control (ethanol treated) and experimental (5-Ph-IAA treated) worms were grown on NGM plates at 20°C and assayed at the young adult

stage. Worms were individually transferred onto an unseeded NGM plate and allowed to recover from transfer for 30 s. An eyelash pick dipped in 100% 1-octanol (Sigma-Aldrich, 472328) was laid perpendicular to the forward-moving worm approximately 3 mm away. Worms were trialed three times each with 10 s of buffer time in between each trial. Worms were scored for backwards movement, or positive avoidance response (1), or lack of an avoidance response (0), and the average avoidance ratio was calculated for each worm. Worms were also scored categorically for average avoidance time (2, 5 or 10 s).

## Microscopy
Worms were anesthetized using 100 mM sodium azide ($NaN_3$) and mounted on 5% agarose pads on glass slides. *z*-stack images (0.5-1 μm thick) were acquired using a Zeiss confocal microscope (LSM980) or Zeiss compound microscope (Imager Z2) with ZEN Blue software. Maximum intensity projections of 2-30 slices were generated with Fiji/ImageJ software (Schindelin et al., 2012).

## Statistical analysis
All statistical analyses were performed on GraphPad Prism 10.

## Acknowledgements
We thank Chi Chen for expert assistance in strain generation, Daniel Shaye for the LiveAct strain, Nikolaos Stefanakis for cell ID, Giulio Valperga for the pan-neuronal TIR1 line used in this study, the knockout group at Tokyo Women's Medical University Hospital for mutant strains and members of the Hobert lab for comments on the manuscript.

## Competing interests
The authors declare no competing or financial interests.

## Author contributions
Conceptualization: O.H.; Formal analysis: A.S., B.V., S.S., D.M.M., G.R.A., Y.H.R.; Funding acquisition: O.H.; Investigation: A.S., B.V., S.S., D.M.M., G.R.A., Y.H.R.; Project administration: O.H.; Supervision: O.H.; Visualization: A.S., B.V., S.S., D.M.M., G.R.A.; Writing – original draft: O.H.; Writing – review & editing: A.S., B.V., S.S., D.M.M., G.R.A., Y.H.R.

## Funding
This work was funded by the National Institute of Neurological Disorders and Stroke (R01NS039996, R01NS137594) and the Howard Hughes Medical Institute. Open Access funding provided by Columbia University. Deposited in PMC for immediate release.

## Data and resource availability
All relevant data and details of resources can be found within the article and its supplementary information.

## The people behind the papers
This article has an associated 'The people behind the papers' interview with some of the authors.

## Peer review history
The peer review history is available online at https://journals.biologists.com/dev/lookup/doi/10.1242/dev.205502.reviewer-comments.pdf

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
