## [Peer Review File · Development (Cambridge, England)]

A conserved *C. elegans* zinc finger-homeodomain protein, ZFH-2, is continuously required for the structural integrity and function of the alimentary tract and gonad

Antoine Sussfeld, Berta Vidal, Surojit Sural, Daniel M. Merritt, G. Robert Aguilar, Yasmin H. Ramadan and Oliver Hobert
DOI: 10.1242/dev.205502

Editor: Swathi Arur

Review timeline

Original submission:	12 January 2026
Editorial decision:	2 February 2026
First revision received:	13 February 2026
Accepted:	18 February 2026

Original submission

First decision letter

MS ID#: dev.205502

MS TITLE: A conserved *C. elegans* zinc finger-homeodomain protein, ZFH-2, continuously required for structural integrity and function of alimentary tract and gonad

AUTHORS: Antoine Sussfeld, Berta Vidal, Surojit Sural, Daniel M. Merritt, Robert Aguilar, Yasmin Ramadan and Oliver Hobert

Dear Dr Hobert,

I have now received all the referees reports on the above manuscript, and have reached a decision. The referees' comments are appended below.

The overall evaluation is positive and we would like to publish a revised manuscript in *Development*. However, each of the two reviewers provides some recommendations to further clarify the manuscript. Please attend to all of the reviewers' comments in your revised manuscript and detail them in your point-by-point response. If you do not agree with any of their criticisms or suggestions explain clearly why this is so.

Reviewer 1

Advance summary and potential significance to field

This paper investigates the *C. elegans* ZFH-2 transcription factor orthologous to mammalian ZFH2-4 transcriptional regulators. The authors report the developmental expression pattern of the protein and show that mutations in the gene result in larval lethality secondary to digestive tract defects. Other defects in the reproductive system are also noted using conditional manipulations of gene function. Structure/function studies suggest that some homeodomains but not the Zn finger domains of the protein may be essential for function.

The ZFH2-4 family of transcription factors is unusual in that the proteins contain a mix of several Zn finger and homeodomains in the context of large proteins containing sometimes over 3000 amino acids. Little is known about the functions of this class of regulators in any system, and this manuscript begins to shed light on their activities using *C. elegans* as a model setting. The experiments in the paper are rigorous and convincing, and the conclusions drawn are reasonable. Although the mechanism of action of ZFH-2 (i.e. target genes) is not systematically explored, the authors do an excellent job characterizing cell-specific effects of the protein and provide a solid framework for future studies. I have no issues with the science. I only have one minor comment:

1. In Figure 2, the expression patterns are a bit difficult to follow. There are many dots of varying sizes, and it is not clear which are nuclei and which are other structures that may be unrelated. Is there a way to make this more clear using color images and not just grey scale images?

Reviewer 2

Advance summary and potential significance to field

This manuscript describes a detailed genetic and expression pattern analysis of the conserved zinc finger homeodomain protein ZFH-2 in *C. elegans*. The *zfh-2* gene and its homologs are relatively poorly studied despite their conservation, and this study provides perhaps the most comprehensive view of both expression patterns and mutant phenotypes for a member of this family in any organism, so is likely to be broadly impactful. To me the most striking finding is the seemingly similar cellular disorganization or adhesion phenotypes in several lumen-containing tissues (pharyngeal valve, rectum, spermatheca, vulva), a finding that is well supported by the organismal phenotype and expression data. The writing is generally clear and supported by the data throughout. I have various minor suggestions but this is overall an excellent study.

Minor suggestions:

1.

The seemingly widespread expression of *zfh-2* in both neuronal and non-neuronal cell types but lack of specification defects is striking. Combined with the organizational defects in the valve, this suggests some role in cell organization that is regulated somewhat independently from fate itself. The expression timing of *zfh-2* (starting during or after terminal differentiation) supports this too. The statement in the second to last paragraph of the discussion that "ZFH-2 may regulate the expression of cell surface molecules required for the proper arrangement of cells into tubular structures" is appropriate. Whether its targets and functions are similar or different in other cell types, especially the nervous system is intriguing as well, and maybe worth a brief discussion here; in general this part of the discussion could be expanded because I think there are fairly broadly interesting implications of these results

2.

Regarding this statement at the end of page 1: "expression outside the nervous system...has not previously been examined." This should be reworded to specify it refers specifically to the protein since RNA expression information in neuronal and non-neuronal cell types is established in many bulk and single cell RNA-seq studies. The Ma et al study also described embryonic expression in several of the nonneuronal cell types where this study finds ZFH-2 in the adult. I mention this not to diminish the current (more comprehensive) expression analysis, but rather because these earlier large scale studies provide additional information about timing of expression and the RNA data supports the accuracy of the current data (using tagged alleles). I'm not sure the best place to discuss this but suggest a slight rewording here and perhaps a few sentences in the discussion.

3.

ZFH3 structure is shown in Fig 1B but ZFH2 homeodomains used in phylogeny in Fig 1C - not sure why (or if it makes any qualitative difference) but was distracting to me.

4.

Figure 3- Are there similar "holes" (vacuolated structures) near the intestinal-rectal valve at this stage? Such defects are identified in older worms later with a weaker allele, but maybe worth mentioning here given that ZFH-2 is expressed in both places.

5.

"defects in pharyngeal muscle differentiation, specifically in pm8 (Fig 3E)" - it is clear pm8 is either not expressing the CEH-22 marker (a differentiation defect as stated) or is in the wrong place (a different type of defect), but hard to tell which, maybe worth clarifying this or rewording. Also, pm8 fate is notable in being induced in late embryos by a Notch signal (see Rasmussen et al 2008 from the Priess lab). Since it appears this is the only cell that is plausibly mis-specified in the zfh-2 mutants, maybe the possibility this could be due to a failure in this induction is worth mentioning?

6.

Does constitutive depletion of degron allele phenocopy the null? (should be possible to test this easily using the embryo-permeable auxin analog 5-phIAA-AM and the same eft-3::TIR1 driver). Strictly optional but would be a very useful piece of data on the strength of knockdown that would provide some context for the postembryonic auxin treated conditions and phenotypes.

7.

Were any valve structural defects seen in the post-embryonic knock down conditions?

8.

"We ascribe this to the additional function of ZFH-2 in vpi cells which may affect pumping behavior as well." - I agree this is the most likely explanation, but without a valve driver maybe a fairer wording would be "This could be due to additional function of ZFH-2 in vpi cells..."

7.

Not a suggestion, just wanted to commend the authors for the detailed structure-function work in Figures 7 and 8.

First revision

Author response to reviewers' comments

RESPONSE TO REVIEWERS COMMENTS (in red)

Reviewer 1: This paper investigates the *C. elegans* ZFH-2 transcription factor orthologous to mammalian ZFH2-4 transcriptional regulators. The authors report the developmental expression pattern of the protein and show that mutations in the gene result in larval lethality secondary to digestive tract defects. Other defects in the reproductive system are also noted using conditional manipulations of gene function. Structure/function studies suggest that some homeodomains but not the Zn finger domains of the protein may be essential for function.

The ZFH2-4 family of transcription factors is unusual in that the proteins contain a mix of several Zn finger and homeodomains in the context of large proteins containing sometimes over 3000 amino acids. Little is known about the functions of this class of regulators in any system, and this manuscript begins to shed light on their activities using *C. elegans* as a model setting. The experiments in the paper are rigorous and convincing, and the conclusions drawn are reasonable. Although the mechanism of action of ZFH-2 (i.e. target genes) is not systematically explored, the authors do an excellent job characterizing cell-specific effects of the protein and provide a solid framework for future studies. I have no issues with the science. I only have one minor comment:

1. In Figure 2, the expression patterns are a bit difficult to follow. There are many dots of varying sizes, and it is not clear which are nuclei and which are other structures that may be unrelated. Is there a way to make this more clear using color images and not just grey scale images?

We have added more labels to the figure and also indicated gut autofluorescence with pink asterisks so that it is not mistaken as real signal. We have decided to keep the grey scale images since swapping to color images would not solve the gut autofluorescence problem. As indicated in the manuscript text we mainly focus on non-neuronal expression in this figure and show a detailed

list of all neuronal expression in Table S1 which had previously been described in Reilly et al, Nature 2020.

Reviewer 2: This manuscript describes a detailed genetic and expression pattern analysis of the conserved zinc finger homeodomain protein ZFH-2 in *C. elegans*. The *zfh-2* gene and its homologs are relatively poorly studied despite their conservation, and this study provides perhaps the most comprehensive view of both expression patterns and mutant phenotypes for a member of this family in any organism, so is likely to be broadly impactful. To me the most striking finding is the seemingly similar cellular disorganization or adhesion phenotypes in several lumen-containing tissues (pharyngeal valve, rectum, spermatheca, vulva), a finding that is well supported by the organismal phenotype and expression data. The writing is generally clear and supported by the data throughout. I have various minor suggestions but this is overall an excellent study.

Minor suggestions:

1. The seemingly widespread expression of *zfh-2* in both neuronal and non-neuronal cell types but lack of specification defects is striking. Combined with the organizational defects in the valve, this suggests some role in cell organization that is regulated somewhat independently from fate itself. The expression timing of *zfh-2* (starting during or after terminal differentiation) supports this too. The statement in the second to last paragraph of the discussion that "ZFH-2 may regulate the expression of cell surface molecules required for the proper arrangement of cells into tubular structures" is appropriate. Whether its targets and functions are similar or different in other cell types, especially the nervous system is intriguing as well, and maybe worth a brief discussion here; in general this part of the discussion could be expanded because I think there are fairly broadly interesting implications of these results.

We appreciate that the reviewer is interested in further discussion and we gladly obliged by adding some more discussion on *zfh-2* function in the nervous system in the Discussion section.

2. Regarding this statement at the end of page 1: "expression outside the nervous system...has not previously been examined." This should be reworded to specify it refers specifically to the protein since RNA expression information in neuronal and non-neuronal cell types is established in many bulk and single cell RNA-seq studies. The Ma et al study also described embryonic expression in several of the nonneuronal cell types where this study finds ZFH-2 in the adult. I mention this not to diminish the current (more comprehensive) expression analysis, but rather because these earlier large scale studies provide additional information about timing of expression and the RNA data supports the accuracy of the current data (using tagged alleles). I'm not sure the best place to discuss this but suggest a slight rewording here and perhaps a few sentences in the discussion.

We fully appreciate this point. We have clarified in the text that "ZFH-2 protein outside of the nervous system [...] has not previously been examined" (page 2) and we have added a sentence in the results section mentioning previous RNA-sequencing studies (page 3).

3. ZFH3 structure is shown in Fig 1B but ZFH2 homeodomains used in phylogeny in Fig 1C - not sure why (or if it makes any qualitative difference) but was distracting to me.

Thanks for catching this. Now including both ZFH2 and 3 (since they both cover the two versions of these proteins, having either 3 or 4 homeodomains).

4. Figure 3- Are there similar "holes" (vacuolated structures) near the intestinal-rectal valve at this stage? Such defects are identified in older worms later with a weaker allele, but maybe worth mentioning here given that ZFH-2 is expressed in both places.

Based on DIC we do not observe obvious defects in intestinal-rectal valve of *zfh-2* null mutants at the L1 stage. Either they are too small to easily observe by light microscopy at this stage, or these defects only appear later on, as a result of continuous pressure/stress in the structure during defecation, as we observe in the HD2 deletion mutant that grows to adulthood.

5. "defects in pharyngeal muscle differentiation, specifically in pm8 (Fig 3E)" - it is clear pm8 is either not expressing the CEH-22 marker (a differentiation defect as stated) or is in the wrong place (a different type of defect), but hard to tell which, maybe worth clarifying this or rewording.

Also, pm8 fate is notable in being induced in late embryos by a Notch signal (see Rasmussen et al 2008 from the Priess lab). Since it appears this is the only cell that is plausibly mis-specified in the *zfh-2* mutants, maybe the possibility this could be due to a failure in this induction is worth mentioning?

First, we would like to clarify that we looked at *myo-2*, not *ceh-22*, marker. Regarding the pm8 differentiation versus position defect, we count one less *myo-2* positive cell within the pharynx which suggests a differentiation defect in pm8. One could consider a scenario in which pm8 is now located outside of the pharynx; unfortunately, the transgene we used to assess *myo-2* expression also contains a *myo-3* reporter in the same array which marks body wall muscle (*oxIs322*). That is the reason there are positive nuclei outside the pharynx in pictures of Fig 3E (we have now clarified this in the Figure legend). We cannot completely rule out the possibility that pm8 may have been extruded from the pharynx in *zfh-2* null mutants and we have added a sentence in the results section to acknowledge this possibility (page 3).

We thank the reviewer for the reminder regarding the Notch signal. The timing of *zfh-2* function argues that it is functioning at least in part after the Notch signal, it is an excellent possibility that *zfh-2* acts to properly interpret the Notch signal. We have now added a discussion of this point in the Discussion section.

6. Does constitutive depletion of degron allele phenocopy the null? (should be possible to test this easily using the embryo-permeable auxin analog 5-phIAA-AM and the same *eft-3::TIR1* driver). Strictly optional but would be a very useful piece of data on the strength of knockdown that would provide some context for the postembryonic auxin treated conditions and phenotypes.

Constitutive depletion of ZFH-2 using AID2 produces a strong loss-of-function phenotype, but it does not fully phenocopy the null. In experiments where *zfh-2::mIAA7* animals were transferred to auxin plates at mid-L4 stage, all of the progeny of these animals arrested during larval development (between L1 and L3 stages), in contrast to the early L1 arrest in *zfh-2* null animals. When *zfh-2::mIAA7* animals were transferred to auxin plates at mid-L1 stage, the phenotypes (pharyngeal pumping, fertility, vpi holes and spermatheca defects) were comparable to those observed in the hypomorphic *zfh-2* allele with deletion of only the 2nd homeodomain. Hence, we conclude that though maternal depletion of ZFH-2 is not sufficient to fully phenocopy the null phenotype in the progeny of those animals, the AID2 approach still gives an opportunity to investigate the postembryonic functions of this protein in the same generation. We felt that this point may not warrant an extensive description in the paper since there are of course technical limitations to these auxin experiment.

7. Were any valve structural defects seen in the post-embryonic knock down conditions?

We appreciate that the reviewer nudged us to look at this. In animals with ZFH-2 depletion from mid-L1, we indeed observed holes in the pharyngeal-intestinal valve region. These images have been added to Fig.5F.

8. "We ascribe this to the additional function of ZFH-2 in vpi cells which may affect pumping behavior as well." - I agree this is the most likely explanation, but without a valve driver maybe a fairer wording would be "This could be due to additional function of ZFH-2 in vpi cells..."

We fully agree with the reviewer. After addition of new data showing holes in the vpi region of ZFH-2 depleted animals, we have changed this sentence to indicate the possibility of ZFH-2 acting in vpi cells to disrupt pumping behavior. The rephrased sentence is: "*In adult animals with ubiquitous ZFH-2 depletion from mid-L1, we observed holes in the vpi region (Fig.5F). This could be due to a continuous requirement of ZFH-2 in maintaining the structural integrity of the pharyngeal-intestinal valve, which may affect pumping behavior as well.*"

9. Not a suggestion, just wanted to commend the authors for the detailed structure-function work in Figures 7 and 8.

Thank you!

Second decision letter

MS ID#: dev.205502R1

MS TITLE: A conserved *C. elegans* zinc finger-homeodomain protein, ZFH-2, continuously required for structural integrity and function of alimentary tract and gonad

AUTHORS: Antoine Sussfeld, Berta Vidal, Surojit Sural, Daniel M. Merritt, Robert Aguilar, Yasmin Ramadan and Oliver Hobert

Dear Dr Hobert,

I am happy to tell you that your manuscript has been accepted for publication in *Development*, pending our standard publication integrity checks.